# Impact of a mobile decision support tool on antimicrobial stewardship indicators in St. John's, Canada

**Daniel Doyle[1], Gerald McDonald[2], Claire Pratt[1], Zahra Rehan[1], Tammy Benteau[3], Jennifer Phillips[2], Peter Daley[1,2]***

1 Memorial University of Newfoundland, St. John's, Newfoundland and Labrador, Canada, 2 Eastern Health Region, St. John's, Newfoundland and Labrador, Canada, 3 Choosing Wisely Newfoundland and Labrador, St. John's, Newfoundland and Labrador, Canada

* pkd336@mun.ca

## Abstract

### Objectives

Inappropriate antibiotic use contributes to antimicrobial resistance. The Spectrum™ app provides antibiotic decision support, based on local antimicrobial resistance rates. We determined the impact of regional implementation of the app on inpatient antimicrobial appropriateness, inpatient antimicrobial usage (AMU), population-based *Clostridioides difficile* infection (CDI) rates and cost, using a retrospective, before and after quasi-experimental design, including a one-year study period.

### Methods

The Spectrum™ app was released to prescribers in February, 2019. We performed two one-day inpatient point prevalence surveys using the National Antimicrobial Prescribing Survey tool, six months before (June 25, 2018) and six months after (June 25, 2019) app dissemination. Inpatient AMU in Defined Daily Dose/1000 patient days and CDI incidence were compared, before and after app dissemination.

### Results

The pre-survey included 184 prescriptions, and the post-survey included 197 prescriptions. Appropriateness was 97/176 (55.1%) pre, and 126/192 (65.6%) post (+10.5%, p = 0.051). Inpatient AMU declined by 6.6 DDD/1000 patient days per month, and CDI declined by 0.3 cases per month. Cost savings associated with reduced AMU were $403.98/bed/year and associated with reduced CDI were $82,078/year.

### Conclusion

We observed improvement in antimicrobial stewardship indicators following Spectrum™ implementation. We cannot determine the cause of these improvements.

**Data Availability Statement:** All relevant data are within the manuscript and its S1 and S2 Datas.

**Funding:** The authors received no specific funding for this work.

**Competing interests:** The authors have declared that no competing interests exist.

## Introduction

The inappropriate use of antimicrobials results in unnecessary exposure of pathogens to antimicrobial agents, resulting in the development of antimicrobial resistance (AMR) over time. AMR is a threat to global health, requiring strategies to optimize antimicrobial usage. Interventions to improve the appropriateness of antimicrobial prescribing are a key component in antimicrobial stewardship (AMS) programs [1]. Appropriate antimicrobial prescribing includes choice of empiric and targeted therapy selection, and optimal dosage and duration of treatment.

Canadian antimicrobial use (AMU) surveillance reports that Newfoundland and Labrador has the highest AMU rate among provinces [2]. Hospital antimicrobial purchasing rate in the Atlantic provinces is twice as high as in Ontario [2].

AMS programs improve patient outcomes and reduce AMR, healthcare-associated infections, and healthcare spending [3]. Computerized decision support tools reduce mortality, decrease AMR and improve adherence to antibiotic guidelines [4]. A mobile decision support application ("app") improved prescriber guideline adherence among patients admitted with respiratory infections in New Zealand by eight percent, compared to no improvement in a control hospital (p = 0.04) [5]. A cluster randomized trial of a computerized clinical decision support tool reported a nine percent improvement in inpatient antimicrobial appropriateness, defined by match between empiric treatment and microbiology results (OR 1.48, 95% confidence interval 0.95–2.29) [6]. Another decision support tool studied in a randomized trial did not reduce mortality, length of stay, or *Clostridiodes difficile* infection (CDI) rate [7].

Spectrum™ (https://spectrum.app) is a mobile app containing antimicrobial prescribing guidelines based on the local antibiogram, antimicrobial and pathogen information, and it advises on management of antimicrobial allergy, prophylaxis, dosing, duration and de-escalation strategies. The app was programmed by the regional AMS committee, using published or institutional guidelines and consensus recommendations. Following implementation of Spectrum™ in an acute care facility in Saskatoon, Canada, there was a 25% reduction in AMU and a 35% reduction in anti-pseudomonal AMU [8]. This is the only previously published report of impact of the Spectrum™ app.

Our objective was to evaluate the impact of the Spectrum™ app on antimicrobial stewardship outcomes, including inpatient AMU, antimicrobial prescribing appropriateness, CDI rate and cost impact of antibiotic use and CDI reduction. Using a before and after retrospective, quasi-experimental design, we hypothesized that the use of the app would be associated with a reduction in AMU, an increase in appropriateness, a decrease in CDI, and cost savings.

## Methods

### Setting

Two tertiary-care academic hospitals located in St. John's, Newfoundland, Canada were included in the study. St. John's is located in Eastern Canada and the metropolitan area has a population of 212,433 [9].

### AMU and antimicrobial cost

Inpatient Acute Care AMU data from Health Sciences Centre (346 beds) and St Clare's Mercy Hospital (205 beds) wards was collected using the Pyxis™ automated dispensing system. Data was collected from January 2019 to March 2020. Defined daily dose (DDD) was calculated using World Health Organization definitions [10]. DDD was reported instead of days of therapy because we did not have access to patient-level data, and to compare with national rates

reported in DDD. AMU data collection occurred over a 15-month period, from January 2019 to March 2020. January 2019 was the first time that complete AMU data was available. Three hospital wards did not have Pyxis™ dispensing systems, so were not included.

Inpatient pharmacy antimicrobial expenditure data was collected from the pharmacy. Cost savings were calculated as change in antimicrobial expenditure over the study period.

## Antimicrobial prescribing appropriateness

Single day point prevalence surveys estimate trends in appropriateness over time [11]. This study method retrospectively analyses all inpatient antibiotic prescriptions received by the inpatient pharmacy on a single day. Two single day point prevalence surveys were conducted over a one-year period. One survey was conducted six months prior to Spectrum™ release (June 25, 2018), and one survey was conducted six months after (June 25, 2019) Spectrum™ dissemination to health care workers in the region.

The Spectrum™ app was made available to the health region for a six-month trial period, and was promoted to users in a medical grand rounds presentation, internal regional email, and through a local news interview (https://www.cbc.ca/news/canada/newfoundland-labrador/prescription-overuse-nl-1.5026919). The Newfoundland and Labrador Pharmacy Board and Newfoundland and Labrador Medical Association also promoted the app.

Chart review was performed on all inpatient antimicrobial prescriptions on the survey dates. Oral and parenteral antimicrobials were included. Antimicrobials prescribed for prophylaxis, used for a non-infectious disease indication, or with insufficient documentation to determine appropriateness were excluded. Antimicrobial treatment prevalence was calculated as the number of patients receiving any antimicrobial treatment on the day of the survey, divided by the number of occupied beds.

Appropriateness of prescription is defined as ideal antimicrobial usage considering the clinical information available. Appropriateness of prescription was assessed using the National Antimicrobial Prescribing Survey (NAPS) tool [12]. NAPS is an Australian quality improvement audit tool, recently promoted in Canada, which includes a more holistic appropriateness assessment than only guideline compliance. The NAPS tool defines appropriateness using a two-category assessment (appropriate or inappropriate) and a four-category assessment (optimal, adequate, suboptimal, inadequate, or unable to determine) for each prescription [13, 14].

Appropriateness was determined subjectively using chart review, and included indication, route of administration, dosage, frequency, duration, dosage adjustment for renal impairment, mismatch between treatment and reported antimicrobial susceptibility, drug allergy history, spectrum of activity for empiric treatment, and adherence to local guidelines where available. (13) Appropriateness assessments were performed by an internal medicine resident and reviewed by an infectious disease physician.

## CDI rate and cost of CDI

CDI was prospectively surveilled, including inpatients and outpatients and reported as cases per 100,000 inhabitants living in the Eastern Health region. Stool testing method was polymerase chain reaction for DNA and toxin. Only toxin positive cases were included. CDI data collection occurred over a 15-month period, from January 2019 to March 2020. The cost of one CDI case was estimated as $11,930 [15].

## Statistical analysis

AMU was calculated monthly as DDD/1000 patient days, using the denominator of number of inpatient beds occupied on the day of survey. Drugs of AMS concern were analysed

individually. Linear trend in monthly AMU and CDI was calculated. Pre and post treatment prevalence was compared using two-sided t test. Appropriateness was compared using both two-category and four-category appropriateness assessments, and pre and post appropriateness were analysed using two-sided Pearson chi-square. Appropriateness was compared in subgroups by drug, drug class, physician specialty, and reason for appropriateness.

### Ethics

The study design was exempted from ethics approval by the Health Research Ethics Board. Participant informed consent was not required and was not collected. Patient identifiers were removed prior to analysis.

## Results

### Uptake

Spectrum™ was accessed 20,016 times between February 1 and June 24, 2019, with a mean of 598 unique monthly active users (range 214–826), including 25% physicians, 16% residents, 14% pharmacists and 8% nurse practitioners. Feedback was passively collected within the app, and was received from 50 users, with positive ratings.

### AMU and antimicrobial cost

Total inpatient AMU rates demonstrated a trend in reduction between January 2019 (560 DDD/1000) and March 2020 (519 DDD/1000) (-41 DDD/1000, -12%, slope of trend line -6.62 DDD/1000/month). See Fig 1.

Antimicrobial categories of AMS all demonstrated a trend in reduction over the study period. The slope of the trend line for piperacillin-tazobactam usage was -0.0607 DDD/1000/month, for fluoroquinolone usage was -0.0389 DDD/1000/month, for vancomycin usage was -0.0191 DDD/1000/month, and for carbapenem usage was -0.0468 DDD/1000/month.

Reduction in AMU over six months resulted in overall antimicrobial expenditure savings of $111,296, or $403.98 per acute care bed per year.

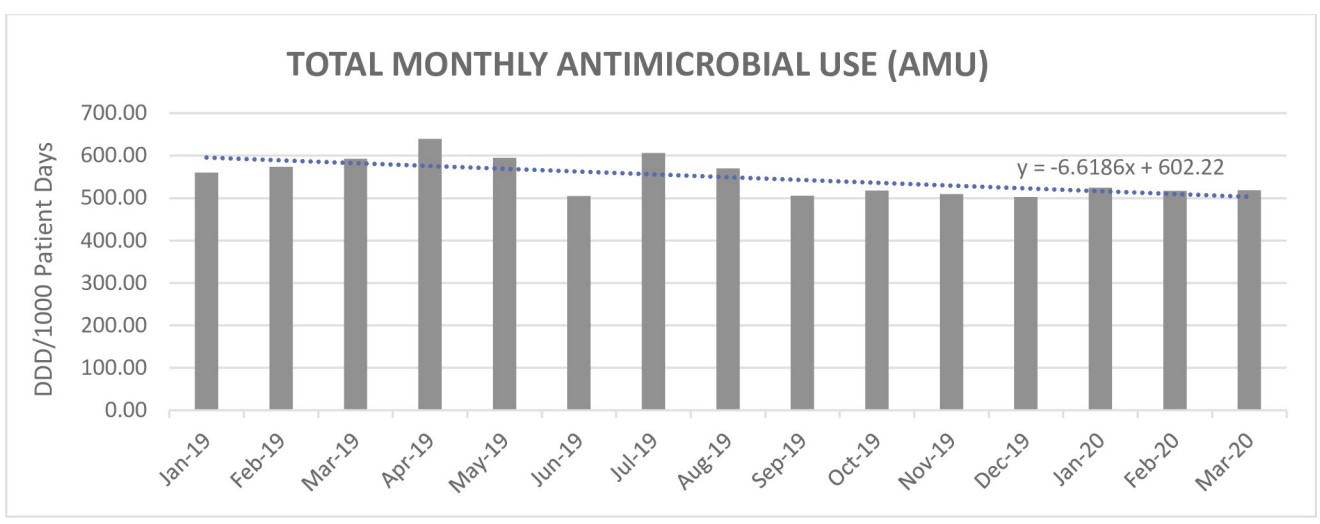

**Fig 1. Total antimicrobial use.**

## Appropriateness

There were 276 antimicrobial prescriptions in the pre-survey chart review. 92 prescriptions (92/276 = 33%) were excluded. There were 282 antimicrobial prescriptions in the post-survey chart review. 85 prescriptions (85/282 = 30%) were excluded. See Fig 2.

Antibiotic treatment prevalence was 131/420 (31.2%) occupied beds pre-survey, and 139/429 (32.4%) occupied beds post-survey (+1.2%, p = 0.70).

Overall antimicrobial prescribing appropriateness was 97/176 (55.1%) in the pre-survey and 126/192 (65.6%) in the post-survey (+10.5%, p = 0.051). By appropriateness category, optimal increased from 50% to 57.3% (+7.3% (p = 0.19)), adequate increased from 5.1% to 8.3% (+3.2% (p = 0.31)), suboptimal decreased from 25.6% to 20.3% (-5.3% (p = 0.28)) and inadequate decreased from 19.3% to 14.1% (-5.2% (p = 0.22)). See Fig 3.

By speciality, internal medicine appropriateness increased from 52.9% to 67.5% (+14.6% (p = 0.10)), and critical care appropriateness increased from 64.7% to 85% (+20.3% (p = 0.17)). Surgical specialty appropriateness decreased from 54.9% to 52.7% (-2.2% (p = 0.91)). See Fig 4.

By drug class, carbapenem appropriateness increased the most, from 16.7% to 85.7% (+69% (p = 0.053)), fluoroquinolone appropriateness increased from 18.2% to 43.5% (+25.3% (p = 0.13)), cefazolin appropriateness increased from 66.7% to 78.9% (+12.2% (p = 0.61)), ceftriaxone appropriateness increased from 69.6% to 83.3% (+13.7% (p = 0.51)), and metronidazole appropriateness increased from 41.7% to 53.8% (+12.1% (p = 0.84)). Piperacillin-tazobactam

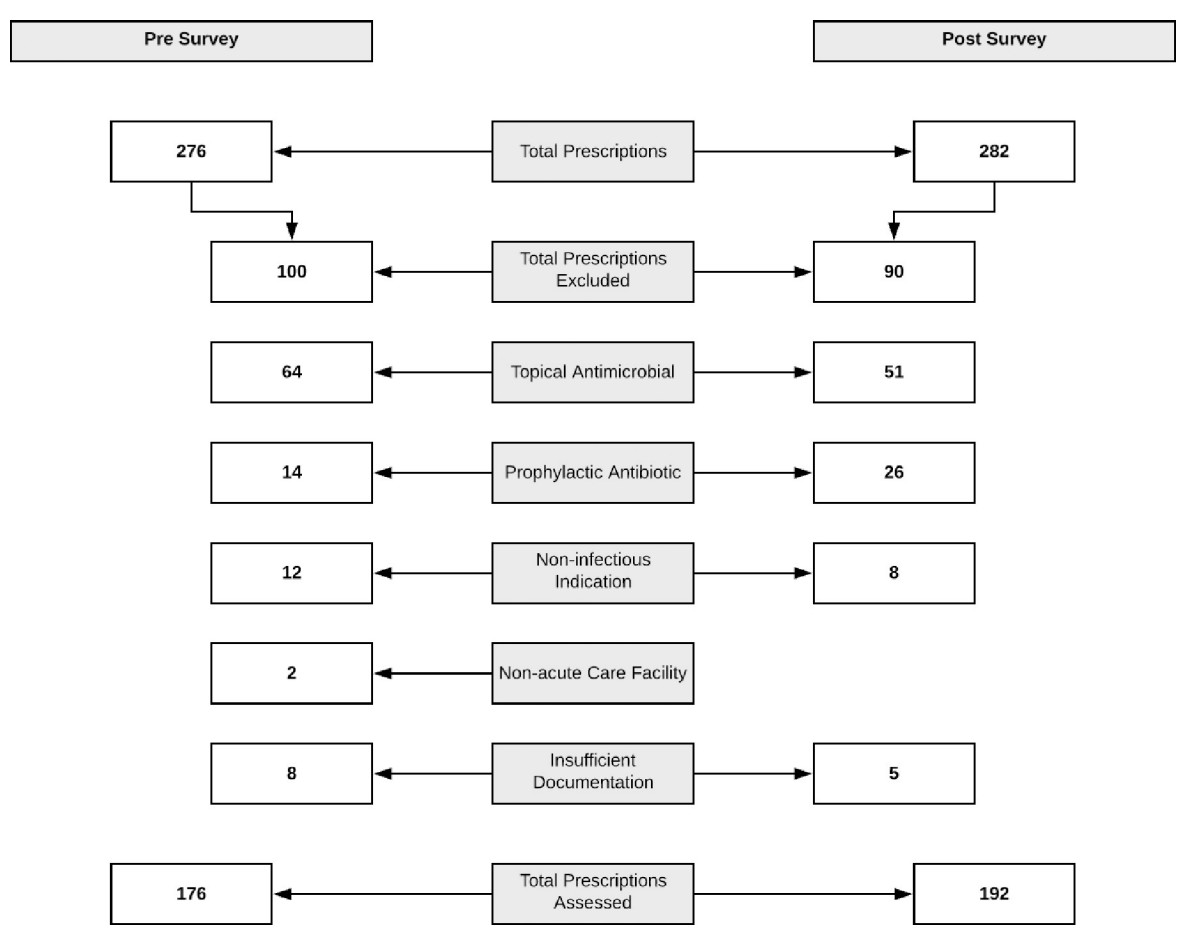

**Fig 2. Prescription inclusion.**

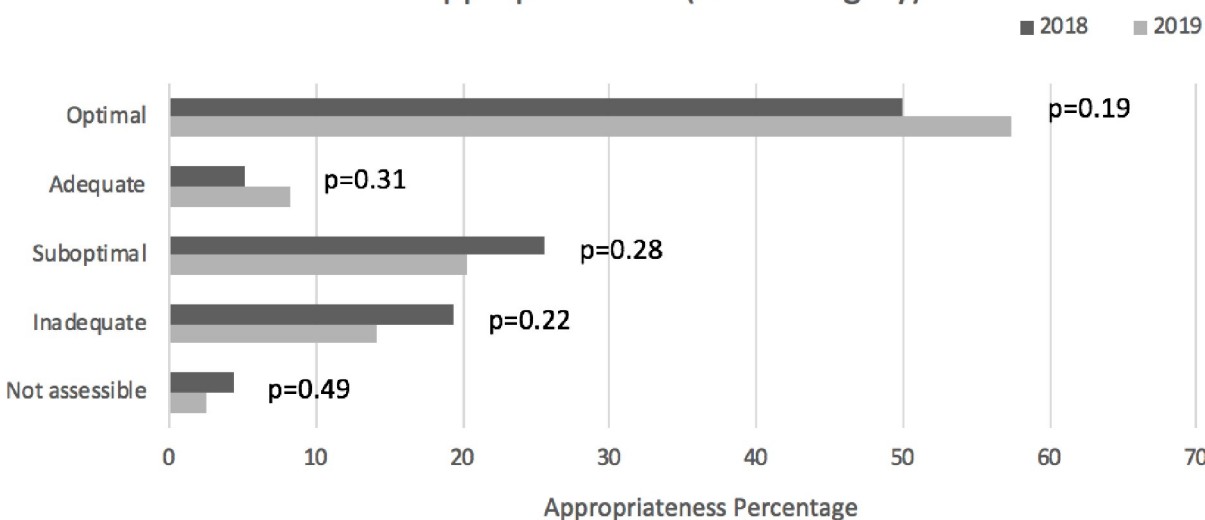

**Fig 3. Antimicrobial prescribing appropriateness (Four category).**

appropriateness decreased from 48.1% to 41.9% (-6.2% (p = 0.83)) and vancomycin appropriateness decreased from 84.2% to 77.3% (-6.9% (p = 0.87)). See Fig 5.

Refer to Table 1 for results of appropriateness by category. Improvements were observed in appropriateness of dose/frequency, duration, route, microbiology mismatch, missing indication, and target tissue mismatch.

Asymptomatic bacteriuria (ASB) treatment as a percentage of prescriptions with no indication decreased from 45.5% to 28.6% (-16.9% (p = 0.51)), and ASB treatment as a percentage of all prescriptions decreased from 5.7% to 2.1% (-3.6% (p = 0.13)).

## CDI and cost of CDI

January 2019 was the first month in which all laboratories used a common stool testing method. CDI incidence declined continuously, from 6.3/100,000 inhabitants in January 2019,

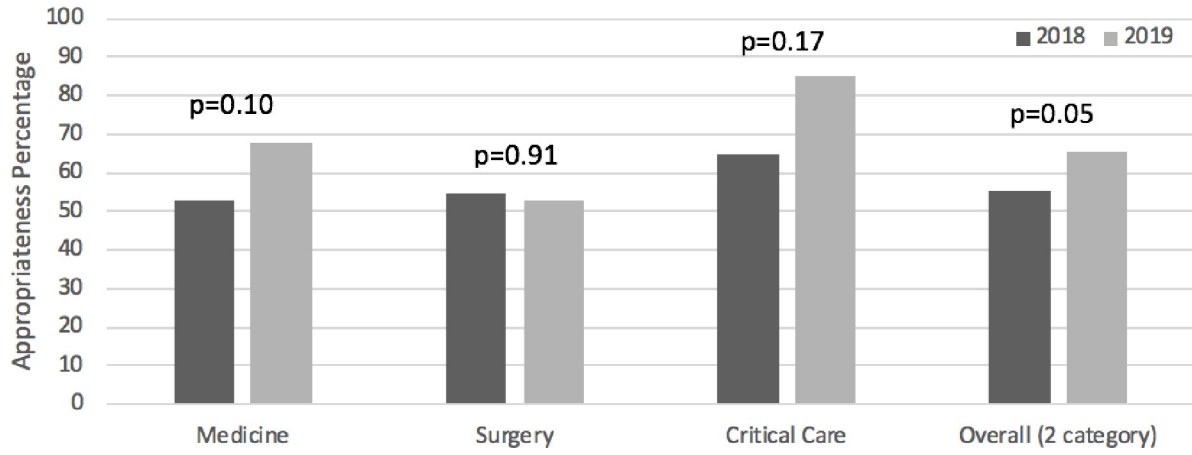

**Fig 4. Antimicrobial prescribing appropriateness by specialty.**

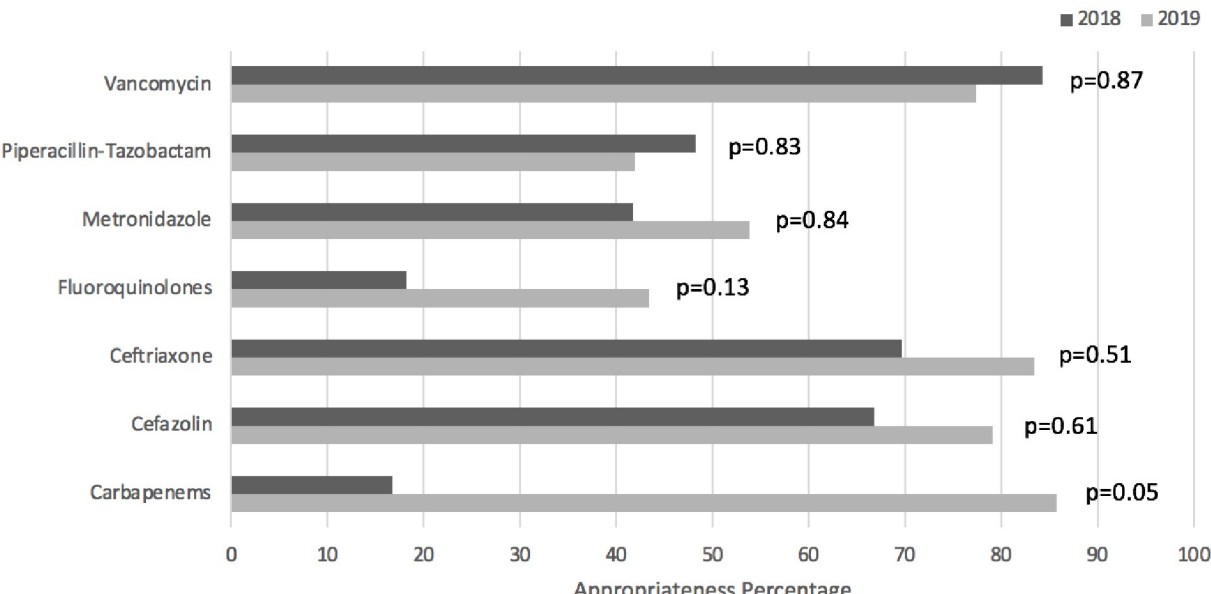

**Fig 5. Antimicrobial prescribing appropriateness by drug class.**

to 4.4/100,000 in March 2020 (-1.9/100,000, -30%, slope of trend line -0.30 cases/month). There was no seasonal variation over 15 months. We did not have access to data on site of acquisition of CDI in hospital or community. Using the trend line, we calculated an absolute reduction of -8.6 CDI cases over 15 months, associated with a savings of $82,078 per year.

## Discussion

Improvements were reported in objective and subjective antimicrobial stewardship indicators during the period of implementation of the app. The app was successfully disseminated, and engagement continues after the study period. Improvements were clinically but not generally statistically significant and were associated with significant cost savings. Improvements were observed in medical and critical care specialties, in almost all antibiotic classes, and in six out

**Table 1. Appropriateness by category.**

| Appropriateness Category Breakdown | | | |
|---|---|---|---|
| | **2018 (%)** | **2019(%)** | **P value** |
| Allergy | 100 | 99.5 | 1.00 |
| Dose/Frequency | 94.9 | 98.4 | 0.10 |
| Duration | 92.1 | 95.8 | 0.19 |
| Micro mismatch | 94.3 | 99.0 | 0.03 |
| No indication | 87.5 | 92.7 | 0.13 |
| Route | 94.3 | 98.4 | 0.06 |
| Spectrum too broad | 93.2 | 87.0 | 0.07 |
| Spectrum too narrow | 98.9 | 98.4 | 1.00 |
| Surgical prophylaxis | 99.4 | 95.3 | 0.04 |
| Tissue Mismatch | 99.4 | 99.5 | 1.00 |

of ten reasons for appropriateness. The inappropriate treatment of ASB, a previously identified local stewardship problem [16] improved during the study period.

We cannot determine the cause of these stewardship improvements. Concurrently, the regional antimicrobial stewardship committee was active in measurement of inpatient AMU and in other AMS interventions (formulary restriction, auto-stop at seven days duration, therapeutic drug monitoring, cascade susceptibility reporting), and these co-interventions may have also improved stewardship indicators. These co-interventions were not initiated or changed during the study period. There were no new infection control interventions initiated or stopped during the study period.

Ours is the first study to report the impact of the Spectrum^TM decision support tool on inpatient appropriateness and population-based CDI rate. We observed reduction in CDI, but this trend has also been observed in national CDI surveillance over the same period. A previous randomized trial of a different decision support intervention did not show reduction in CDI [7], suggesting that our observed CDI reduction may not have been caused by our intervention.

Inpatient AMU is reported in Canada by the Canadian Antimicrobial Resistance Surveillance Program(2), which includes only purchasing-level data for hospitals (doses purchased, not doses dispensed), so we did not compare our AMU results to this report. The Canadian Nosocomial Infection Surveillance Program (CNISP) reports doses dispensed by inpatient bed-days [17]. Their most recent report from 2016 demonstrated a national inpatient AMU of 555 DDD/1000, which is comparable to our initial finding of 580.4 DDD/1000, and greater than our final result of 513.7 DDD/1000 (-59.3 DDD/1000, -10.3%), suggesting that our two hospitals are similar in usage to national AMU rates. Inpatient AMU in the Eastern Canadian region in 2016 (not including the two hospitals in our study) was 453 DDD/1000, suggesting that our two hospitals have a higher AMU rate than other hospitals in our region.

Our results reveal AMS problems in our setting. We observed a reduction in appropriateness of piperacillin-tazobactam prescriptions during the study period. Patients were frequently under-dosed for infections requiring empiric coverage of *Pseudomonas aeruginosa* [18]. Also, piperacillin-tazobactam was prescribed empirically for bacterial meningitis on two occasions, but this drug does not cross the blood-brain-barrier.

Treatment of ASB detected on routine pre-operative urine cultures was responsible for 10/176 (5.7%) of inpatient antimicrobial prescriptions in the pre-survey, and 4/192 (2.1%) of inpatient antimicrobial prescriptions in the post-survey (-3.6%, p = 0.071). This treatment is not associated with reduction in operative infections after prosthetic joint surgery and is inappropriate [19, 20].

The use of broad-spectrum agents in the setting of uncomplicated cholecystitis is another potential target for stewardship intervention, as this was the surgical indication most commonly treated with broader coverage than required.

There was also a trend toward greater appropriateness amongst internal medicine and critical care services. This trend was not observed amongst surgical specialties. Certain surgical sub-specialties do not have resident physicians, and it is possible that these services had less uptake of Spectrum^TM use. One study showed that younger prescribers, defined as being less than 44 years of age, were 17.3% more likely to use mobile applications compared to their older colleagues (p<0.05) [21].

Future research based on our results could include a cluster randomized trial of the Spectrum^TM decision support tool, in order to control for the impact of co-interventions. Decision support tools are a single component of multi-intervention AMS programs, and are implemented in combination with prescriber education, restrictive policies and prospective audit and feedback. Further interventions targeting our identified AMS problems could include

prospective audit and feedback of broad-spectrum antimicrobials, and restricted access to urine culture.

## Limitations

Our study design cannot infer causation, but only correlation. Appropriateness assessment based on retrospective chart review may be biased by the quality of medical records, or the opinion of the assessor. The relatively small sample size may have reduced the likelihood of finding statistically significant differences in subgroup analysis categories, while exaggerating differences amongst other categories. In addition, the study being conducted over two single-day surveys gives a very brief overview of prescribing practices on a select day. Based on our study design, day-to-day variability in our institutions prescribing appropriateness would not be captured in our analysis.

## Conclusions

We observed improvement in AMS indicators during the period of implementation of the Spectrum™ decision support app. We cannot conclude that the intervention caused the improvements.

## Supporting information

**S1 Data. Appropriateness assessment data.**
(XLSX)

**S2 Data. Inpatient AMU Jan 2019-Mar 2020.**
(XLSX)

## Acknowledgments

The Spectrum™ app was provided by the Spectrum company on a six-month free trial.

This manuscript was presented as two oral abstracts at the online Association of Medical Microbiology and Infectious Diseases Canada meeting, June 2020.

## Author Contributions

**Conceptualization:** Peter Daley.

**Investigation:** Daniel Doyle, Gerald McDonald, Claire Pratt, Zahra Rehan, Tammy Benteau, Jennifer Phillips.

**Project administration:** Peter Daley.

**Supervision:** Peter Daley.

**Visualization:** Daniel Doyle.

**Writing – original draft:** Daniel Doyle.

**Writing – review & editing:** Gerald McDonald, Claire Pratt, Zahra Rehan, Tammy Benteau, Jennifer Phillips, Peter Daley.

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
