## [Decision Letter · Decision Letter 0]

25 Nov 2020

PONE-D-20-30999

Impact of a Mobile Decision Support Tool on Antimicrobial Stewardship Indicators in St. John’s, Canada

PLOS ONE

Dear Dr. Daley,

Thank you for submitting your manuscript to PLOS ONE. After careful consideration, we feel that it has merit but does not fully meet PLOS ONE’s publication criteria as it currently stands. Therefore, we invite you to submit a revised version of the manuscript that addresses the points raised during the review process.

We look forward to receiving your revised manuscript.

Kind regards,

Mehreen Arshad, M.D.

Academic Editor

PLOS ONE

Journal Requirements:

3. Please upload a copy of Supporting Information Figures S1 - S5 and Supporting Information Table S1 which you refer to in your text on page 23.

Reviewers' comments:

Reviewer's Responses to Questions

**Comments to the Author**

1. Is the manuscript technically sound, and do the data support the conclusions?

Reviewer #1: Yes

Reviewer #2: Partly

2. Has the statistical analysis been performed appropriately and rigorously? 

Reviewer #1: Yes

Reviewer #2: No

3. Have the authors made all data underlying the findings in their manuscript fully available?

Reviewer #1: Yes

Reviewer #2: No

4. Is the manuscript presented in an intelligible fashion and written in standard English?

Reviewer #1: Yes

Reviewer #2: Yes

5. Review Comments to the Author

Reviewer #1: The manuscript entitled, “Impact of a mobile decision support tool on antimicrobial stewardship indicators in St. John’s, Canada” evaluated the impact of a decision support tool had on antibiotic utilization and incidence of Clostridium difficile after dissemination of the clinical decision support tool. This topic is clinically relevant and incorporation of clinical decision support tools is recommended within antimicrobial stewardship guidelines. The co-investigators detail how clinicians were informed of the clinical decision support tool and utilized a standardize method to determine appropriateness of antimicrobial use. However, the co-investigators did not pursue an explanation as to why antimicrobial use was measured by defined daily dose as oppose to days of therapy. Also, the co-investigators did not comment on other antimicrobial stewardship efforts at the two academic hospitals that could affect antimicrobial use and incidence of Clostridium difficile, and the seasonal variations in Clostridium difficile. In addition, even though the overall appropriate use of antimicrobials increased and was found to be statistically significant, when these data are further refined by specialty, drug class, and category of appropriateness, the differences were clinically significant but were not found to be statistically significant. Without further details on other antimicrobial stewardship efforts at the two academic hospitals, it is difficult to attribute these changes solely to the clinical decision tool. Therefore, content of the manuscript would require further clarification (please see comments to be addressed by the authors).

The following are comments to be addressed by the authors:

Introduction:

Page 3, line 50: I was unable to retrieve the content of this website. May consider updating link or if available, cite another source.

Page 3, line 59: Suggest change “Odds ratio” to “OR”.

Methods:

Would consider adding other antimicrobial stewardship efforts by the two academic hospitals involved in study. This would allow the reader to determine established stewardship initiatives at these institutions or if there were other new stewardship efforts that were implemented at the time of incorporating the clinical decision tool.

AMU:

Page 4, line 84: What was defined daily dose (DDD) used to determine antimicrobial use as oppose to days of therapy (DOT)? Based on the Infectious Diseases Society of American guidelines, DOT is recommended to monitor antimicrobial use and DDD is an alternative if patient-level antibiotic use data is not available.

Appropriateness:

Good explanation of methodology used to determine appropriateness.

CDI:

Page 6, lines 120-121: Would consider moving this statement to discussion section.

Results-uptake:

Page7, line 137: Was feedback actively pursued? Based on the number of unique monthly active users, this is a limited number of users that provided feedback.

Page 7, lines 148-149: Consider changing “Piperacillin-tazobactam” and “Vancomycin” to “piperacillin-tazobactam” and “vancomycin”.

CDI:

As there are seasonal variations in Clostridium difficile with an increased incidence during winter months (reference: Gilca R, Fortin E, Frenette C, Longtin Y, Gourdeau M. Antimicrobial Agents and Chemotherapy 2012:56(2): 639-646), would suggest commenting on if the time of year may have an impact on the incidence of CDI. In addition, would suggest commenting on if there were changes in the incidence of hospital-acquire CDI at the study hospitals or if there were other infection control efforts that may have impacted the incidence of CDI during the study period.

Discussion:

Would consider commenting on results of a randomized study that evaluated the impact of a clinical decision support tool on clinical outcomes (i.e., length of stay, 30-day mortality, 30-day re-admission) and C. difficile infection, acquisition of multidrug resistant organisms, and antibiotic cost (reference: Ridgway JP, Robicsek A, Shah N, et al. Clinical Infectious Diseases 2020).

Table 1:

The table is a duplication of text. Would consider omission of text and reference to results in Table 1.

Reviewer #2: Abstract

1. Recommend to add the background of study with the problem statement in the abstract section.

2. Suggest further description on the methods in the abstract section. The statement of "We performed two one-day inpatient point prevalence surveys using the National Antimicrobial Prescribing Survey tool, before (June 25, 2018) and six months after (June 25, 2019) app dissemination to prescribers" is unclear. When is exactly the implementation date of the SpectrumTM and how long is the study period?

3. Recommend the authors to provide a careful study conclusion especially by inferring the direct impact/causal relationship of SpectrumTM in improving AMS indicators (in the abstract section).

Please refer the attachment for more comments.

6. PLOS authors have the option to publish the peer review history of their article (what does this mean?). If published, this will include your full peer review and any attached files.

Reviewer #1: No

Reviewer #2: No

---

## [Author Response · Author response to Decision Letter 0]

8 Jan 2021

Public Library of Science

Response to Reviewers

Impact of a Mobile Decision Support Tool on Antimicrobial Stewardship Indicators in St. John’s, Canada

January 6, 2021

Dear Editor,

Thanks for your thoughtful peer review on our submitted article. We respond with a revised version, including response to each suggestion below.

We have ensured that our manuscript matches the journal style requirements, and provided additional details regarding participant consent and anonymization of participant data prior to analysis, in the manuscript and online submission. Supporting information figures 1-5 and table 1 have been uploaded in .tif format. We have also uploaded the raw data. 

Response to Reviewer comments:

Reviewer #1: The manuscript entitled, “Impact of a mobile decision support tool on antimicrobial stewardship indicators in St. John’s, Canada” evaluated the impact of a decision support tool had on antibiotic utilization and incidence of Clostridium difficile after dissemination of the clinical decision support tool. This topic is clinically relevant and incorporation of clinical decision support tools is recommended within antimicrobial stewardship guidelines. The co-investigators detail how clinicians were informed of the clinical decision support tool and utilized a standardize method to determine appropriateness of antimicrobial use. However, the co-investigators did not pursue an explanation as to why antimicrobial use was measured by defined daily dose as oppose to days of therapy. Also, the co-investigators did not comment on other antimicrobial stewardship efforts at the two academic hospitals that could affect antimicrobial use and incidence of Clostridium difficile, and the seasonal variations in Clostridium difficile. In addition, even though the overall appropriate use of antimicrobials increased and was found to be statistically significant, when these data are further refined by specialty, drug class, and category of appropriateness, the differences were clinically significant but were not found to be statistically significant. Without further details on other antimicrobial stewardship efforts at the two academic hospitals, it is difficult to attribute these changes solely to the clinical decision tool. Therefore, content of the manuscript would require further clarification (please see comments to be addressed by the authors).

The following are comments to be addressed by the authors:

Introduction:

Page 3, line 50: I was unable to retrieve the content of this website. May consider updating link or if available, cite another source.

Thanks, we have updated the link. 

Page 3, line 59: Suggest change “Odds ratio” to “OR”.

Thanks, we have revised this.

Methods:

Would consider adding other antimicrobial stewardship efforts by the two academic hospitals involved in study. This would allow the reader to determine established stewardship initiatives at these institutions or if there were other new stewardship efforts that were implemented at the time of incorporating the clinical decision tool.

Thanks, we described AMS co-interventions in the discussion section, but have added further detail on ongoing AMS efforts.

AMU:

Page 4, line 84: What was defined daily dose (DDD) used to determine antimicrobial use as oppose to days of therapy (DOT)? Based on the Infectious Diseases Society of American guidelines, DOT is recommended to monitor antimicrobial use and DDD is an alternative if patient-level antibiotic use data is not available.

Thanks, DDD was used because we do not have access to patient-level data. This has been added to the manuscript. 

Appropriateness:

Good explanation of methodology used to determine appropriateness.

CDI:

Page 6, lines 120-121: Would consider moving this statement to discussion section.

Results-uptake:

Page 7, line 137: Was feedback actively pursued? Based on the number of unique monthly active users, this is a limited number of users that provided feedback.

Page 7, lines 148-149: Consider changing “Piperacillin-tazobactam” and “Vancomycin” to “piperacillin-tazobactam” and “vancomycin”.

Thanks, we have moved the statement to the discussion. User feedback was passively received, and feedback uptake was low. We changed to lower case on drug names. 

CDI:

As there are seasonal variations in Clostridium difficile with an increased incidence during winter months (reference: Gilca R, Fortin E, Frenette C, Longtin Y, Gourdeau M. Antimicrobial Agents and Chemotherapy 2012:56(2): 639-646), would suggest commenting on if the time of year may have an impact on the incidence of CDI. In addition, would suggest commenting on if there were changes in the incidence of hospital-acquire CDI at the study hospitals or if there were other infection control efforts that may have impacted the incidence of CDI during the study period.

Thanks, we only have access to 15 months of population-level CDI incidence data, demonstrating reduction over that time. We did not observe seasonal variations during that period. We don’t have hospital-acquired or community-acquired CDI breakdown. There were no changes made in ongoing infection control interventions during the study. 

Discussion:

Would consider commenting on results of a randomized study that evaluated the impact of a clinical decision support tool on clinical outcomes (i.e., length of stay, 30-day mortality, 30-day re-admission) and C. difficile infection, acquisition of multidrug resistant organisms, and antibiotic cost (reference: Ridgway JP, Robicsek A, Shah N, et al. Clinical Infectious Diseases 2020).

Thanks, we have added this reference.

Table 1:

The table is a duplication of text. Would consider omission of text and reference to results in Table 1.

Thanks, we have minimized the text description.

Reviewer #2: Abstract

1. Recommend to add the background of study with the problem statement in the abstract section.

Thanks, we have added this to the abstract.

2. Suggest further description on the methods in the abstract section. The

statement of "We performed two one-day inpatient point prevalence surveys

using the National Antimicrobial Prescribing Survey tool, before (June 25, 2018)

and six months after (June 25, 2019) app dissemination to prescribers" is

unclear. When is exactly the implementation date of the SpectrumTM and how

long is the study period?

Thanks, we have added the implementation date and the study period.

3. Recommend the authors to provide a careful study conclusion especially by

inferring the direct impact/causal relationship of SpectrumTM in improving AMS

indicators (in the abstract section).

Thanks, we modified the conclusion wording. 

Introduction

1. Recommend further explanation and elaboration on: AMU, AMR, AMS,

SpectrumTM and the outcome measures ie. appropriateness of antimicrobial

prescribing, AMU and CDI. Although the authors had explained on the

SpectrumTM, it would be better if they could also explain on the previous

evaluation or validation conducted on the apps.

Thanks, we have added further elaboration in the Introduction section. We have included the only previous publication evaluating the Spectrum app. 

2. Page 3, line 54-56 - Recommend the authors to revise this statement: "A

mobile decision support application (“app”) improved guideline adherence

among patients admitted with respiratory infections in New Zealand by eight

percent, compared to no improvement in a control hospital (p=0.04)". Based on

the reference cited, the mobile apps improved guideline adherence by the

prescriber, but not the guideline adherence among patients...

Thanks, we have revised this.

4. Page 4, line 72 - The objectives statement is unclear especially for ".....

and inpatient appropriateness....." - suggest to revise.

Thanks, we have revised this.

Methods

x1. Page 4, line 78 - Recommend to describe clearly on the study setting. What do

you mean by "both tertiary-care academic hospitals....."?

Thanks, we have added description to the study setting.

2. Kindly clarify the duration of study: 1 year (25 June 2018 to 25 June 2019 OR

January 2019 to March 2020)?

Thanks, we have clarified study period.

3. Recommend the authors to revise the methods section to increase clarity and

allow reproducibility.

Thanks, we have added clarity regarding the single day point prevalence survey method.\\

4. Suggest the authors to revise the subtopic for the methods. eg. Appropriateness - In what aspect/scope?

Thanks, we have revised the subtitle and added clarity on the definition of appropriateness used. 

5. Recommend to further describe the statistical analysis and need to correspond

to the results section.

Thanks, we have added detail to the statistical analysis section and rearranged this section to match the results section. 

Results

1. Recommend further description on the costing in methods and data/analysis

in the manuscript.

Thanks, we have added the source of the drug expenditure data. 

2. Suggested to revise the statements on the appropriateness (page 14, line 156-

158).

Thanks, lines 156-158 describe the inclusion of prescriptions in the study. 

3. Recommend further elaboration/description for statement ; page 14, line 162-

163.

Thanks, we have added clarity to antimicrobial treatment prevalence in the methods section.

4. Suggest the author to improve the presentation of results. It would be better

not in a form of graph or bar chart only, but in a form of table might further

increase readers understanding. The current bar charts are not self-explanatory.

There are quite number of missing details/information.

Thanks, the figure titles are included separately, which may add clarity to the figures. 

5. Suggest the author to improve Table 1 - How did you get the p value ie. using

what statistical test.

Thanks, the statistical methods have been clarified in the methods section.

6. Overall, recommend the author to revise the results section of the manuscript

to increase readers' understanding.

Thanks, we have made small changes in the results section to improve understanding.

Discussion

1. Suggested that the Discussion section contain specific recommendations for future

research to improve AMS outcomes.

Thanks, we have added this.

2. Recommend the authors to elaborate on the inpatient bed-days vs. inhabitant-days

and purchasing-level data. Please explain the concerns in detailed. How do they affect

the findings?

Thanks, these terms have been removed. 

3. Page 17, line 231 - 235 - suggest the authors to discuss more on the statements.

Thanks, we have added content on the comparison between our AMU rate and national AMU rate reports.

4. Discussions were more on the appropriateness of antimicrobial prescribing.

Thanks, we have added discussion on all results.

5. Recommend to discuss also on the utilization and CDI.

Thanks, we have added discussion on AMU and CDI.

6. Suggest the authors to add conclusion section.

Thanks, we have added a conclusion section.

---

## [Decision Letter · Decision Letter 1]

12 Feb 2021

PONE-D-20-30999R1

Impact of a Mobile Decision Support Tool on Antimicrobial Stewardship Indicators in St. John’s, Canada

PLOS ONE

Dear Dr. Daley,

Thank you for submitting your manuscript to PLOS ONE. After careful consideration, we feel that it has merit but does not fully meet PLOS ONE’s publication criteria as it currently stands. Therefore, we invite you to submit a revised version of the manuscript that addresses the points raised during the review process.

We look forward to receiving your revised manuscript.

Kind regards,

Mehreen Arshad, M.D.

Academic Editor

PLOS ONE

Reviewers' comments:

Reviewer's Responses to Questions

**Comments to the Author**

1. If the authors have adequately addressed your comments raised in a previous round of review and you feel that this manuscript is now acceptable for publication, you may indicate that here to bypass the “Comments to the Author” section, enter your conflict of interest statement in the “Confidential to Editor” section, and submit your "Accept" recommendation.

Reviewer #2: (No Response)

2. Is the manuscript technically sound, and do the data support the conclusions?

Reviewer #2: Yes

3. Has the statistical analysis been performed appropriately and rigorously? 

Reviewer #2: Yes

4. Have the authors made all data underlying the findings in their manuscript fully available?

Reviewer #2: Yes

5. Is the manuscript presented in an intelligible fashion and written in standard English?

Reviewer #2: Yes

6. Review Comments to the Author

Reviewer #2: Most of the concerns raised by the reviewers were well and properly addressed by the authors. Thank you. However, there are still minor amendments required as suggested below:

1. There are information on the cost saving from the implementation of the SpectrumTM app, although it was not stated in the objective of the study. There is no details on how the costing were done or calculated. Since the study objectives did not include cost, the authors may just omit the component in the manuscript.

2. Please address the reviewer suggestion to include in the discussion section: "Would consider commenting on results of a randomized study that evaluated the impact of a clinical decision support tool on clinical outcomes (i.e., length of stay, 30- day mortality, 30-day re-admission) and C. difficile infection, acquisition of multidrug resistant organisms, and antibiotic cost (reference: Ridgway JP, Robicsek A, Shah N, et al. Clinical Infectious Diseases 2020).

7. PLOS authors have the option to publish the peer review history of their article (what does this mean?). If published, this will include your full peer review and any attached files.

Reviewer #2: No

---

## [Author Response · Author response to Decision Letter 1]

29 Apr 2021

Dear Editor,

Thanks for your thoughtful peer review on our submitted article. We respond with a revised version, including response to each suggestion below.

Response to Reviewer comments:

Reviewer #2: Most of the concerns raised by the reviewers were well and properly addressed by the authors. Thank you. However, there are still minor amendments required as suggested below:

1. There are information on the cost saving from the implementation of the SpectrumTM app, although it was not stated in the objective of the study. There is no details on how the costing were done or calculated. Since the study objectives did not include cost, the authors may just omit the component in the manuscript.

Thanks, we have added cost analysis to the objectives of the study, and further information on cost analysis methods. 

2. Please address the reviewer suggestion to include in the discussion section: "Would consider commenting on results of a randomized study that evaluated the impact of a clinical decision support tool on clinical outcomes (i.e., length of stay, 30- day mortality, 30-day re-admission) and C. difficile infection, acquisition of multidrug resistant organisms, and antibiotic cost (reference: Ridgway JP, Robicsek A, Shah N, et al. Clinical Infectious Diseases 2020).

Thanks for this suggestion, we had included this reference in the introduction section and have now added discussion comparing our results to this reference. This study used a different decision support intervention and did not demonstrate reduction in CDI.

---

## [Editor Report · Decision Letter 2]

17 May 2021

Impact of a Mobile Decision Support Tool on Antimicrobial Stewardship Indicators in St. John’s, Canada

PONE-D-20-30999R2

Dear Dr. Daley,

We’re pleased to inform you that your manuscript has been judged scientifically suitable for publication and will be formally accepted for publication once it meets all outstanding technical requirements.

Kind regards,

Mehreen Arshad, M.D.

Academic Editor

PLOS ONE

---

## [Editor Report · Acceptance letter]

8 Jun 2021

PONE-D-20-30999R2 

Impact of a Mobile Decision Support Tool on Antimicrobial Stewardship Indicators in St. John’s, Canada 

Dear Dr. Daley:

I'm pleased to inform you that your manuscript has been deemed suitable for publication in PLOS ONE. Congratulations! Your manuscript is now with our production department. 

Kind regards, 

on behalf of

Dr. Mehreen Arshad 

Academic Editor

PLOS ONE